# Scrutinizing Pork Price Volatility in the European Union over the Last Decade

**DOI:** 10.3390/ani12010100

**Published:** 2022-01-01

**Authors:** Katarzyna Utnik-Banaś, Tomasz Schwarz, Elzbieta Jadwiga Szymanska, Pawel Mieczyslaw Bartlewski, Łukasz Satoła

**Affiliations:** 1Department of Management and Economics of Enterprises, University of Agriculture in Kraków, 21 Mickiewicza Avenue, 31-120 Kraków, Poland; lukasz.satola@urk.edu.pl; 2Department of Animal Genetics, Breeding and Ethology, University of Agriculture in Kraków, 24/28 Mickiewicza Avenue, 30-059 Kraków, Poland; tomasz.schwarz@urk.edu.pl; 3Department of Logistics, Institute of Economics and Finance, Warsaw University of Life Sciences—SGGW, 166 Nowoursynowska Street, 02-787 Warsaw, Poland; elzbieta_szymanska@sggw.edu.pl; 4Department of Biomedical Sciences, Ontario Veterinary College, University of Guelph, 50 Stone Road, Guelph, ON N1G 2W1, Canada; pmbart@uoguelph.ca

**Keywords:** pork, market, price, seasonality, cyclical fluctuations, time series

## Abstract

**Simple Summary:**

The volatility of prices for agricultural products is of great importance for both producers (due mainly to continued optimization of the production process) and consumers. We analyzed long-term trends as well as cyclical, seasonal, and irregular fluctuations in pork prices in European Union (EU) countries. Despite the existence of the EU common market, there is considerable spatial variation in pork prices between member states. Pork prices in the EU are characterized by seasonality and are generally higher than average in summer and lower in winter. Considering risk management in agribusiness, the nature of price fluctuations, their amplitude, and period are of paramount importance. Regular seasonal changes or long-term trends can be considered during the decision-making process, but short-term and randomly occurring fluctuations and medium-term changes with large deviations from the expected prices constitute a serious risk.

**Abstract:**

The aim of this study was to analyze the factors that can influence pork prices, particularly the effects of various types of fluctuations on the volatility of pork prices in the European Union as a whole market and individual EU countries. The research material consisted of monthly time series of pork prices collected from 2009 to 2020. These data originated from the Integrated System of Agricultural Information coordinated by the Polish Ministry of Agriculture. Information on global pork production volumes is from the Food and Agriculture Organization Statistics (FAOSTAT) database. Time series of prices were described by the multiplicative model, and seasonal breakdown was performed using the Census X-11 method. The separation of the cyclical component of the trend was performed using the Hodrick–Prescott filter. In 2019, pork production in the European Union totaled 23,954 thousand tonnes, which accounted for 21.8% of global pork production. The largest producers were Germany, Spain, and France, supplying more than half of the pork to the entire European Union market. Pork prices in the EU, averaged over the 2009–2020 period were Euro (EUR) 154.63/100 kg. The highest prices for pork were recorded in Malta, Cyprus, Bulgaria, and Greece, whereas the lowest prices in Belgium, the Netherlands, Denmark, and France. The breakdown of the time series for pork prices confirmed that, in the period from 2009 to 2020, pork prices exhibited considerable fluctuations of both a long-term and medium-term nature as well as short-term seasonal and irregular fluctuations. Prices were higher than average in summer (with a peak in June–August) and lower in winter (January–March). Overall, the proportions of different types of changes in pork prices were as follows: random changes—7.9%, seasonal changes—36.6%, and cyclical changes—55.5%.

## 1. Introduction

Price volatility is an effect of market processes dictating financial changes in both the supply and demand sector. In agriculture, considerable price volatility is due mainly to relatively large price elasticity within the supply chain. Price volatility can also impinge on several market-related variables by directly affecting the cost of production and product storage [1].

The prices of agricultural products are of paramount importance for both the agricultural sector (they drive the optimization of production systems and influence their objectives) and for consumers purchasing specific products. The analysis of market prices also serves as an indirect means of assessing market efficiency. The volatility of prices in European food markets has increased over the past decade, placing agribusinesses at financial risk and uncertainty [2]. Price volatility is a key aspect of financial risk for all market stakeholders, including producers, chains of processing enterprises, and consumers [3].

Not every price change is an indication of economic risk. Recurrent seasonal changes can normally be considered during the process of estimating production profitability. Alternatively, the unexpected and difficult-to-explain price shifts should always be treated as a sign of risk. Understanding the mechanisms influencing price volatility and the ability to break down the general price volatility into seasonal, cyclical and irregular fluctuations can greatly facilitate price risk management. The knowledge surrounding variability in pork prices is of paramount importance due to the significant share of pork (more than one-third) in global meat consumption. Pork production in the EU constitutes more than 20% of the world’s pork production [4].

Price volatility of agricultural products has been addressed in several publications. Utnik-Banaś analyzed volatility of turkey [5], broiler chicken [6], and goose meat prices [7]. Boroumand et al. [8] analyzed the occurrence of “price spikes” of agricultural products and indicated that, since the 1990s, trading of agricultural commodities has been steadily increasing. Empirical evidence suggests that agri-commodities can exhibit sudden and unexpected price shifts. Čechura and Šobrová in 2008 [9] analyzed the price transmission (i.e., the process whereby upstream prices affect downstream prices) in the pork agri-food chain and confirmed the existence of a long-term relationship of the agricultural prices with wholesale prices of processed pig products. It was also concluded that the pork agri-food chain could be characterized as demand driven. Hayes et al. [10] stated that seasonal price changes can result from changes in the supply of pork, consumer demand changes for pork products and a combination of these factors. Apart from these classic factors, there are also other factors that are incidental, especially in periods of major market turmoil. An example is the Coronavirus (COVID-19) pandemic, which has caused supply and demand shocks on an unprecedented scale. Goodwin and Harper [11] examined price interrelationships among pork farming, wholesale, and retail markets in the United States of America, indicating that the US pork sector has undergone many significant structural changes, which may have influenced the price dynamics and transmission of “shocks” through marketing channels. This transmission appears to be largely unidirectional, with the information flowing from farms to wholesale and then to retail markets. Miller and Hayenga [12] also assessed the US pork market and concluded that retail price changes were significantly asymmetric due to low-frequency (slow moving) cycles in wholesale pork prices. Serra et al. [13] critically evaluated price transmission processes in the European Union pork market after the implementation of the single EU market policy in 1993. They found out that pork prices were transmitted across spatially separated EU pig markets and provided evidence for the occurrence of asymmetric (national or regional) price adjustments. Szymańska [14] analyzed the development of the global pork market, emphasizing the progressing globalization, and pointing out that the increasing demand for food combined with the high cost of production was a global challenge for pork producers. Competitiveness of the international pork trade is constantly growing, driven mainly by North and South American countries that can offer an apparent price advantage in exports. The integration of the Finnish meat market (beef and pork) with the EU market was investigated by Liu [15], and the results indicated that meat prices in Finland paralleled those in Germany, but this relationship is only symmetric for pork prices, while it remains asymmetric for beef prices. Price transmission in the Finnish pork market is smoother and more efficient than in the beef market. However, the rate of transmission is still slow compared, for example, with the Danish and German markets. Price dynamics and transmission in the pork markets were also studied by Bakucs et al. [16], Abdulai [17], Babula and Miljkovic [18], Hamulczuk [19,20], Holst and Cramon-Taubadel [21], and Xu et al. [22]. Havlíček et al. [23] analyzed the efficiency of pig production on the international scale and stated that half of the monitored EU countries were ranked as “full-efficiency producers”.

Recently, a new method of big data (BD) analysis was introduced in the food supply chain with pricing, product promotion, product development, and demand forecasting. Subsequently, companies collected a massive amount of data to derive real-time business awareness related to consumers, risk, return, efficiency, and output. Such methods were used by Jagtap and Duong for a food company, with the obtained results having a significant implication for analyzing price fluctuations [24]. An outbreak of African Swine Fever (ASF) in Western European countries strongly influenced the pork industry. It significantly impinged on the economic status of all affected farms and, wherever the transport of pork was banned or restricted, led to the pork market disruptions across the countries [25,26]. In addition, abrupt fluctuations in meat prices, in particular, the occurrence of a significant discrepancy between livestock and wholesale meat prices, were associated with the COVID-19-related economy lockdown [27].

Price volatility is inevitable; however, it is imperative to understand the causes underlying the variability as they may help predict and/or prevent drastic changes in prices. One of the main features of price variability in agriculture includes seasonal or less frequent periodic fluctuations or price cycles. Hence, the aim of this study was to analyze the influence of various fluctuations on the range of pork price volatility in the European Union as a whole (unified) market as well as in individual states within the EU. The newness of the present analysis lies in the consistency of the methodology applied to assess the time series for pork prices. The present results are hence relevant for pork producers as well as policy makers concerned with meat production in the EU. The body of this paper is organized as follows: first, the principles of data analyses are detailed, and then the identified fluctuations in the pork production cycles and prices are presented.

## 2. Materials and Methods

### 2.1. Data

The present research utilized monthly time series for nominal prices of pork in European Union countries, available at the Integrated System of Agricultural Market Information [28]. Basic descriptive statistics of average pork prices in the EU are summarized in Table 1. A plot depicting the lowest and highest pork prices in EU countries is given in Figure 1. Information on world pork production volumes (by country) for the period from 1961 to 2019 is from the FAOSTAT database [4].

### 2.2. Breakdown of Time Series

Time series of pork prices were described using the following multiplicative model representing a product of the identified component [28]:Y_t_ = T_t_ · C_t_ · S_t_ · I_t_
where Y_t_—price in period t, T_t_—long-term trend, C_t_—cyclical fluctuations, S_t_—seasonal fluctuations, and I_t_—irregular fluctuations. The constituent components of the time series were determined with the Census X-11 method [29], using Statistica package 13.1. Seasonality was eliminated from the original series by dividing the empirical price values by the corresponding seasonality coefficients. The significance of seasonal fluctuations was evaluated using the *F* test—a trend cycle extracted from the time series as a Henderson mean. Subsequently, (I) was obtained by dividing the seasonally adjusted time series by the trend-cycle (TC).

The separation of the cyclical component from the trend was accomplished using the Hodrick–Prescott filter to isolate a stochastic smoothly varying trend [30,31]. In the Hodrick–Prescott method, the value of the time series is defined as a sum of a long-term trend and a cyclical component:X_t_ = T_t_ + C_t_
where X_t_—value of the time series, T_t_—value of the long-term trend, and C_t_—value of the cyclical component.

The smoothing parameter was set to a level of λ = 14,400, as monthly data were used. To determine the effects of different types of fluctuations studied on the overall price variability, the share of their variances in the overall variance was determined for different periods (one to twelve months), and mean values for each year were calculated.

Cycles in the analyzed periods were identified using upper and lower turning points as the maximum (peak) and minimum (nadir) values of the model price levels, respectively. To determine whether the directional price changes were random or exhibited a long-term pattern (giving rise to the beginning of a new cycle), respective values for the quarters of cyclical dominance (QCD) were computed; QCD depicts the number of quarters after which the variation in the random component is equal to the variation in the trend component and long-term fluctuations.

## 3. Results and Discussion

### 3.1. Global Pork Production

Global meat production continues to grow steadily. In 1961, world meat production amounted to 71.36 million tonnes, of which 40.3% was beef, 34.7% pork, 12.5% poultry and 12.5% other species (Figure 2).

Over the ensuing six decades, overall meat production increased nearly five-fold to 336.6 million tonnes in 2019. The largest increase was observed in poultry meat production, which increased almost fifteen-fold (to 131.6 million tonnes) between 1961 and 2019. The share of poultry meat increased to 39.1%, with the share of beef declining to 21.6%, while the pork consumption as a % of total meat consumed gradually increased to 39.5% in 1998, and then declined to 32.7% in 2019.

China was the largest pork producer in the world in 2019, with 42.55 million tonnes in 2019, accounting for 38.6% of global production (Table 2). The United States was second and Germany was the third largest pork producer, with 12.54 million tonnes (11.4%) and 5.23 million tonnes (4.8%), respectively. Pork production in these three countries accounted for 55% of global production in 2019. Significant shares of world pork production (over 2%) were also held by Spain (4.2%), Brazil (2.6%), Russia (3.6%), and Vietnam (3.0%).

In the European Union, the volume of pork production in 2019 totaled 23,954 tonnes (Table 3), which accounted for 21.8% of global pork production (FAO). The largest producers were Germany (21.8%), Spain (19.4%) and France (9.2%), supplying more than half of the pork produced in the entire European Union.

### 3.2. Average Pork Prices and Price Fluctuations in the EU

Pork price in the EU, averaged over the period of 2009–2020, was EUR 154.63/100 kg (Figure 3) [26]. The highest prices were recorded in Malta (EUR 208/100 kg) followed by Cyprus (EUR 182), Bulgaria (EUR 181), and Greece (EUR 179). The lowest prices were in Belgium (EUR 137), Netherlands (EUR 140), Denmark (EUR 144), and France (EUR 145).

The breakdown of the time series of prices confirmed that, in the period from 2009 to 2020, pork prices were subject to considerable fluctuations, with a long-term trend and medium-term cyclical fluctuations as well as short-term seasonal and irregular fluctuations (Figure 4 and Figure 5) [28]. The long-term trend is that the average nominal price of pork in the European Union increased slightly from EUR 137 to 160/100 kg. There were three detectable cycles: first from 2009:12 to 2016:2, second up until 2018:6, and then the third ending in 2020:12. Irregular fluctuations in pork prices were characterized by relatively low amplitude, usually not exceeding 2%, but in five instances, they caused monthly price increases by 6% and once a price decrease by 6%.

The occurrence of seasonal influences on pork prices in individual EU countries is shown in Table 4. High values of the *F* test for the EU price time series (32.4) confirm statistically significant seasonal variations in average pork prices on the EU market. The most pronounced seasonal price changes occurred in Spain (F = 52.4), Portugal (*F* = 42.4), Greece (*F* = 35.4), and Czechia (*F* = 32.0). In general, higher prices were recorded in summer months and lower in winter months; the range of seasonal variation was 10.9%, with the highest prices in August (105.1%) and the lowest in January (94.2%; Table 4). The highest amplitude of price shifts in the countries with substantial seasonal price variation was noted in Spain (18.7%), Romania (18.4%), and Portugal (18.2%). These seasonal trends in pork prices result mainly from greater pork production during the winter compared with summer months [32].

On a monthly basis, the seasonal, irregular, and cyclical fluctuations accounted for 48%, 30%, and 22% of the overall variability in pork prices, respectively (Table 5). The half-yearly periods were dominated by cyclical fluctuations (51%), with seasonal and irregular fluctuations amounting to 45% and 4% of detected fluctuations, respectively. On an annual scale, the cyclical, seasonal, and irregular fluctuations accounted for 55%, 37%, and 8% of all variations, respectively.

Other researchers also analyzed the prices of pork in the EU. Hoste [25] pointed out that, at the end of 2019, market prices were high, exceeding EUR 2/kg carcass weight for slaughtered pigs. The global pork production volumes declined by ~25% during that period, leading to high pork prices, especially in China. Industry estimates and model calculations suggest that elevated prices will continue to be seen over the next several years. However, the current market situation is unique, and any further developments can hardly be prognosticated with accuracy.

Fousekis [33] investigated the price dynamics of pork and poultry meat in order to determine whether there is “a single or multiple markets within the EU”. Overall, the empirical results indicate that the EU prices for pork and poultry are heterogeneous. However, Szymańska [34,35] concluded that prices for live pigs in Poland were based on EU prices.

One of the factors that contributed to the pork price increase was improving the welfare of fatteners. Morover, Öhlund et al. [36] discussed negative environmental effects of industrial meat production in Sweden. Environmentally friendly strategies entail sustainable pig farming with a decreased total production of pork and lower emissions of greenhouse gasses per land unit. However, a decrease in production volumes and meat supply to the markets would likely result in another price increase. Western European producers are leaders in implementing sustainability and animal welfare, which reflects the high expectations of domestic consumers and ethical concerns of European citizens [25].

## 4. Conclusions

Despite the existence of an integrated market within the European Union, there is considerable spatial variation in pork prices between member states. During the period from 2009 to 2020, the highest prices of pork were recorded in Malta, Cyprus, Bulgaria and Greece, and the lowest prices in Belgium, Holland, Denmark, and France. Pork prices in the EU are also characterized by seasonality; the prices are higher in summer (with a peak in June–August) and lower in winter (January–March).

The breakdown of the time series for pork prices using the Census II/X11 method allowed us to separate the general price variability into regularly occurring consecutive seasonal fluctuations, cyclical fluctuations lasting several years (with the beginning and end of the cycle rather difficult to determine), and irregular difficult to predict or explain fluctuations.

Certain components of the time series can be used to prognosticate changes in pork prices and implement risk management strategies. In particular, the type of fluctuations as well as their amplitude and time span are important for proper and effective risk management. Regular seasonal changes or long-term trends can be considered during the decision-making process. Alternatively, all the short-term and random fluctuations as well as medium-term changes with large deviations from the expected price level constitute a potential unexpected risk.

## Figures and Tables

**Figure 1 animals-12-00100-f001:**
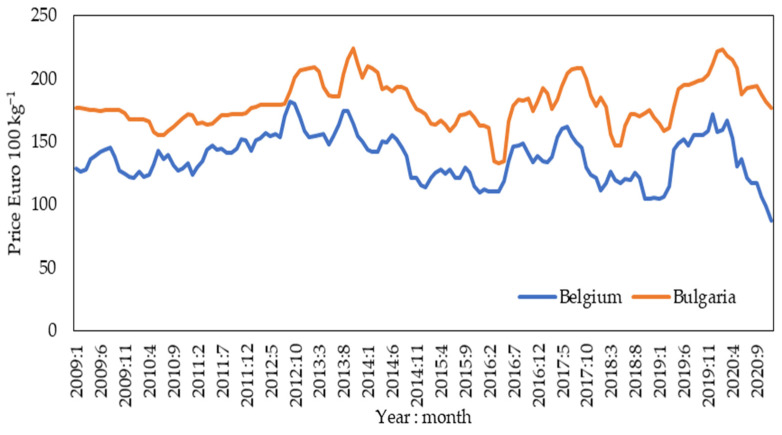
Time series of pork meat prices in Belgium and Bulgaria from 2009 to 2020.

**Figure 2 animals-12-00100-f002:**
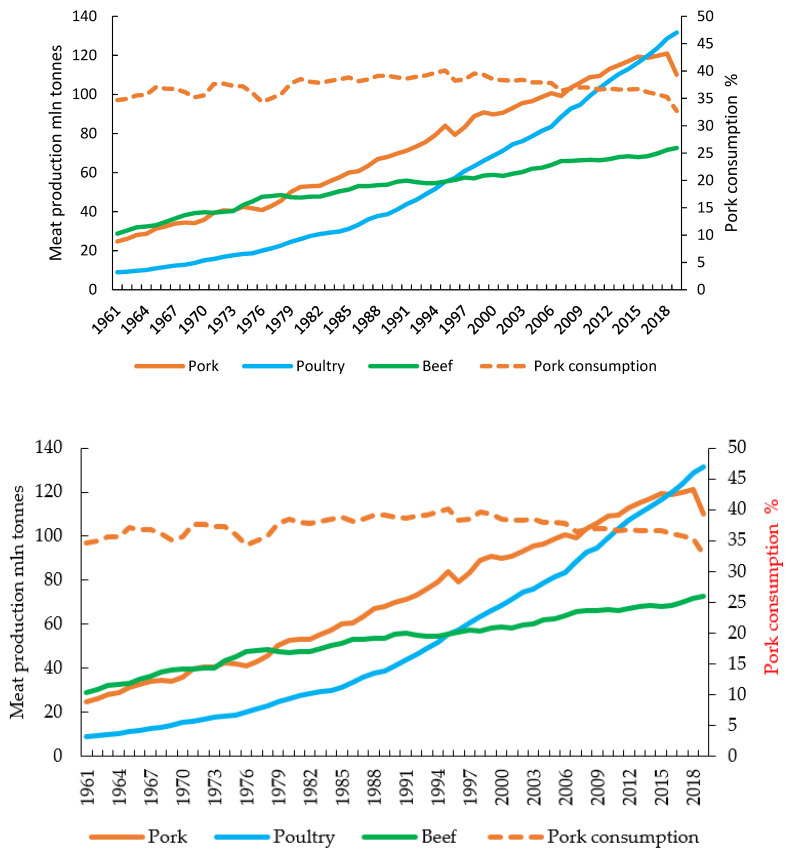
Global meat production (left axis) and pork consumption as a percentage of total meat consumed (right axis) from 1961 to 2019 as based on the FAOSTAT database [4].

**Figure 3 animals-12-00100-f003:**
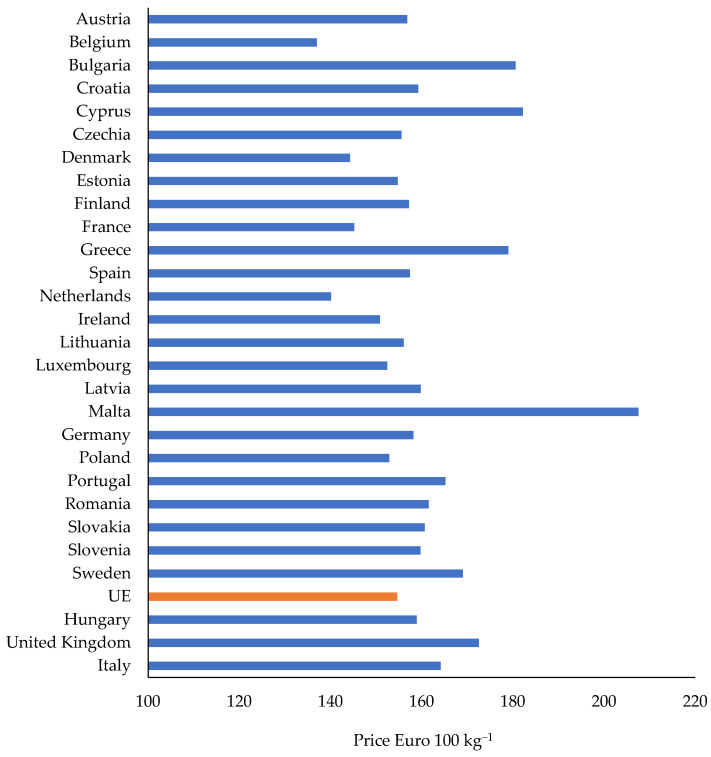
Average pork prices in European Union countries during the period of 2009–2020 (Integrated System of Agricultural Market Information [28]).

**Figure 4 animals-12-00100-f004:**
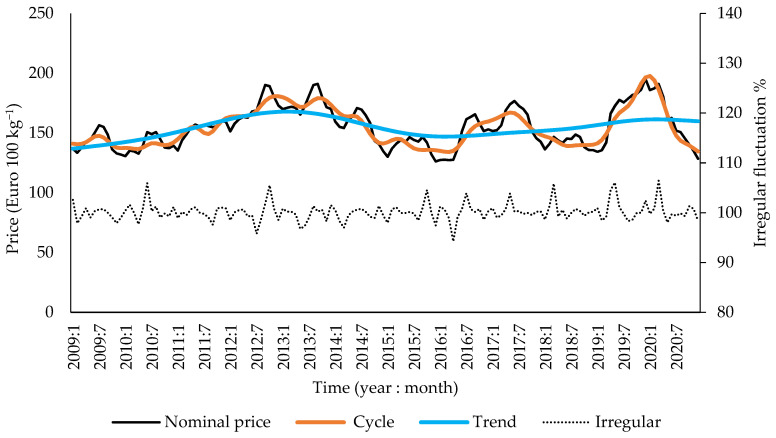
Breakdown of the time series for average pork prices in EU from 2009 to 2020. Left axis units are for nominal prices, cyclical fluctuations, and a long-term trend, and right axis units are for irregular fluctuations.

**Figure 5 animals-12-00100-f005:**
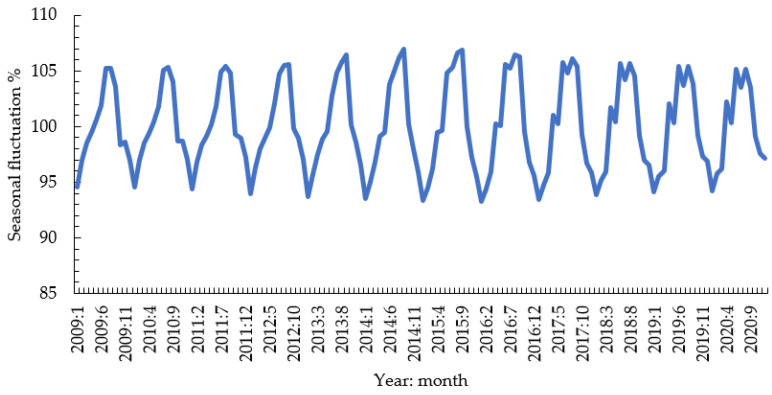
Seasonal fluctuations in average pork prices in the EU from 2009 to 2020.

**Table 1 animals-12-00100-t001:** Descriptive statistics for pork prices in the European Union from 2009 to 2020.

Statistics	Value	Statistics	Value
Mean	154.63	Variability coefficient	0.11
Median	151.83	Skewness	0.4049
Minimum	126.22	Kurtosis	−0.7445
Maximum	195.15	5% percentile	130.26
Standard deviation	17.131	95% percentile	187.10

**Table 2 animals-12-00100-t002:** Top pork producers in the world in 2019.

Country	Production (Millions of Tonnes)	%	Country	Production (Millions of Tonnes)	%
China	42.55	38.6	Mexico	1.60	1.5
USA	12.54	11.4	Denmark	1.50	1.4
Germany	5.23	4.8	Italy	1.46	1.3
Spain	4.64	4.2	Republic of Korea	1.36	1.2
Brazil	4.13	3.7	Japan	1.28	1.2
Russia	3.94	3.6	Myanmar	1.23	1.1
Viet Nam	3.33	3.0	Belgium	1.04	0.9
France	2.20	2.0	Great Britain	0.96	0.9
Canada	2.18	2.0	Thailand	0.94	0.9
Poland	1.99	1.8	Taiwan	0.82	0.7
Philippines	1.84	1.7	Ukraine	0.71	0.6
Netherlands	1.63	1.5	Argentina	0.63	0.6
World	110.11	100.00			

**Table 3 animals-12-00100-t003:** Pork production in the European Union countries in 2019.

Country	Production (Thousands of Tonnes)	Share (%)	Country	Production (Thousands of Tonnes)	Share (%)
Germany	5232.00	21.84	Sweden	240.29	1.00
Spain	4641.16	19.37	Czechia	218.61	0.91
France	2200.35	9.19	Finland	168.95	0.71
Poland	1988.84	8.30	Croatia	120.80	0.50
Netherlands	1628.29	6.80	Greece	86.20	0.36
Denmark	1500.40	6.26	Bulgaria	81.59	0.34
Italy	1464.49	6.11	Lithuania	77.66	0.32
Belgium	1038.99	4.34	Slovakia	72.35	0.30
United Kingdom	960.00	4.01	Estonia	45.29	0.19
Austria	502.03	2.10	Cyprus	43.35	0.18
Hungary	462.06	1.93	Latvia	40.68	0.17
Romania	398.73	1.66	Slovenia	32.21	0.13
Portugal	387.92	1.62	Luxembourg	12.96	0.05
Ireland	304.37	1.27	Malta	4.41	0.02
UE	22,714.83	100.00			

**Table 4 animals-12-00100-t004:** Seasonal fluctuations in pork prices in EU countries.

Country	*F* Test *	Max	Min	Country	*F* Test	Max	Min
Austria	17.2	105.2; VI	93.5; I	Ireland	20.9	102.7; VII	97.1; II
Belgium	16.4	107.6 VI	92.2; I	Latvia	12.9	107.1; IX	93.3; II
Bulgaria	8.0	103.4; VIII	92.8; III	Lithuania	12.1	106.7; VI	93.4; I
Cyprus	15.1	107.4 VIII	92.9; I	Netherlands	16.6	105.5; VI	93.4; I
Czechia	32.0	104.7; VIII	95.5; II	Poland	22.1	106.9; VI	92.5; I
Denmark	18.9	104.3; VI	95.0; III	Portugal	42.4	108.1 VIII	89.9 I
Estonia	24.4	102.9; VI	96.9; III	Romania	21.0	106.8; VIII	88.4 II
Finland	4.5	101.0; XII	99.4; IX	Slovakia	22.6	105.2; VIII	93.6; III
France	21.3	105.5; IX	93.8; II	Slovenia	14.6	105.4; IX	95.4; I
Germany	18.2	106.2; VI	94.3; I	Spain	51.5	108.8; VIII	90.1; I
Greece	35.2	103.9; VIII	93.9 IV	Sweden	9.5	101.5; XII	98.5; III
Hungary	20.9	105.7; VIII	94.8; I	United Kingdom	12.6	101.3 VII	96.9; III
UE	32.4	105.1; VIII	94.2; I				

* *F* test value is significant on a level of *p* < 0.001 for all countries.

**Table 5 animals-12-00100-t005:** Proportions of irregular, cyclical, and seasonal fluctuations in an overall pork price variability in EU countries from 2009 to 2020.

Months	Fluctuations (%)		
	Irregular	Cyclical	Seasonal
1	30.2	22.0	47.8
2	16.3	32.4	51.3
3	8.5	36.9	54.6
4	5.0	40.6	54.4
5	4.3	46.0	49.7
6	3.6	51.2	45.3
7	3.1	58.2	38.7
9	2.7	76.4	20.9
11	1.8	94.4	3.8
12	3.1	96.9	0
Mean	7.9	55.5	36.6

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
