# Peer review of "Scrutinizing Pork Price Volatility in the European Union over the Last Decade"

_animals, 2022, doi:10.3390/ani12010100_

Round 1

Reviewer 1 Report

  • the literature review cites many relevant articles but could be strengthened with including additional work by US authors such as Lee Schultz at Iowa State. Attached is an extension publication talking about seasonality.
  • https://www.extension.iastate.edu/agdm/livestock/html/b2-14.html
  • https://www.meatinstitute.org/ht/a/GetDocumentAction/i/178679
  • there is no discussion introducing Tables 1 and 5
  • Figures 3,4,5 are not referenced in the main body of the text
  • line 119 - space between Yt is needed and the description - price in period t
  • Figure 2 - the second Y axis is titled share of pork (%); I think this should be labelled - Pork Consumption as a % of Total Meat Consumed. This change will also need to be made in line 156
  • lines 234, 235 and 236 are broad statements and need be expanded upon to support the thought or provide some evidence to back-up the claim that producers should target high-end markets for consumers. This may not be more profitable given the extra costs associated with animal welfare and sustainability. 
  • overall, really like the paper - great question and logical display of results.  

Author Response

We would like to begin by thanking the Reviewers and for their suggestions on the article. Our revisions considered all Reviewers’ remarks, which allowed us to significantly improve the manuscript. Detailed responses to Reviewers are given below.

Responses to Reviewer 1

  • the literature review cites many relevant articles but could be strengthened by including additional work by US authors such as Lee Schultz at Iowa State. Attached is an extension publication talking about seasonality.

The suggested items of literature have been cited.

  • there is no discussion introducing Tables 1 and 5

The discussion of Tables 1 and 5 has been added/expanded.

  • Figures 3,4,5 are not referenced in the main body of the text

All figures are also quoted in the text of the manuscript.

  • line 119 - space between Yt is needed and the description - price in period t

It is corrected

  • Figure 2 - the second Y-axis is titled share of pork (%); I think this should be labeled - Pork Consumption as a % of Total Meat Consumed. This change will also need to be made in line 156

Description of the second Y-axis was improved according to the Reviewer suggestion

  • lines 234, 235, and 236 are broad statements and need to be expanded upon to support the thought or provide some evidence to back up the claim that producers should target high-end markets for consumers. This may not be more profitable given the extra costs associated with animal welfare and sustainability. 

The text on this line has been removed, it was not directly related to the topic and requires further research in another publication. We also made an attempt to provide further rationale or explanations in response to the comments above.

Reviewer 2 Report

The authors need to define why pork prices are so important when there is a trend towards vegan and vegetarian foods.

Introduction section needs to be restructured. First paragraph can state what is the issue, second paragraph on why it is an issue, third on what has been done in the past or currently to address this issue and finally the structure of the paper.

The authors have made numerous statements such as pork prices are higher in summer but with no reasoning at all.

Conclusion section needs to be restructured. The bullet points can be part of the conclusion.

Overall, the authors need to clarify what is the novelty in this research. The pork prices data will change in the next few years based on eating habits (religion, veganism, culture), diseases (swine flu, covid-19 etc.), deforestation, climate change, increasing population, etc.  

Many organizations including governmental agencies are using Industry 4.0 technologies such as Internet of things (IoT), Big Data, Blockchain and Cloud analytics to enhance real-time decision-making, transparency, efficiency (Please see the paper- Improving the new product development using big data: A case study of a food company) and these technologies are already addressing the issues and reasons behind price fluctuations of commodities.

Author Response

We would like to begin by thanking the Reviewers and for their suggestions on the article. Our revisions considered all Reviewers’ remarks, which allowed us to significantly improve the manuscript. Detailed responses to Reviewers are given below.

Responses to Reviewer 2

The authors need to define why pork prices are so important when there is a trend towards vegan and vegetarian foods.

To address this issue, we re-emphasized the importance of pork production and sales (e.g., L66-69).

Introduction section needs to be restructured. First paragraph can state what is the issue, second paragraph on why it is an issue, third on what has been done in the past or currently to address this issue and finally the structure of the paper.

We re-organized this section taking into account all of the Reviewer’s suggestions and as indicated by the highlighted sections of our revised manuscript.

The authors have made numerous statements such as pork prices are higher in summer but with no reasoning at all.

Although our primary goals deviated from analyzing the causes of pork price volatility, but rather focused on identifying the types of price fluctuations, we responded to this criticism by elaborating on some putative mechanisms of price shifts ( e.g., L79-84, L250-252).

Conclusion section needs to be restructured. The bullet points can be part of the conclusion.

Albeit not in the bullet form, we have now revised and partially rewritten the concluding statements.

Overall, the authors need to clarify what is the novelty in this research. The pork prices data will change in the next few years based on eating habits (religion, veganism, culture), diseases (swine flu, covid-19 etc.), deforestation, climate change, increasing population, etc.

Please see L61-69 and L124-129.

Round 2

Reviewer 2 Report

Dear Authors

you have not addressed one of the comments from my previous review report

"Many organizations including governmental agencies are using Industry 4.0 technologies such as Internet of things (IoT), Big Data, Blockchain and Cloud analytics to enhance real-time decision-making, transparency, efficiency (Please see the paper- Improving the new product development using big data: A case study of a food company) and these technologies are already addressing the issues and reasons behind price fluctuations of commodities."

Also, you need to correct the reference list 8a (Hayes et al.) and Boroumand et al.

thanks 

Author Response

Responses to the Reviewer 2:

"Many organizations including governmental agencies are using Industry 4.0 technologies such as Internet of things (IoT), Big Data, Blockchain and Cloud analytics to enhance real-time decision-making, transparency, efficiency (Please see the paper- Improving the new product development using big data: A case study of a food company) and these technologies are already addressing the issues and reasons behind price fluctuations of commodities."

It has been done on the text in lines 112-117.

Also, you need to correct reference list 8a (Hayes et al.) and Boroumand et al.

The literature was corrected.
